# Association of COVID-19 Vaccines ChAdOx1-S and BNT162b2 with Circulating Levels of Coagulation Factors and Antithrombin

**DOI:** 10.3390/vaccines10081226

**Published:** 2022-07-31

**Authors:** Amal Hasan, Hossein Arefanian, Arshad Mohamed Channanath, Irina AlKhairi, Preethi Cherian, Sriraman Devarajan, Thangavel Alphonse Thanaraj, Mohamed Abu-Farha, Jehad Abubaker, Fahd Al-Mulla

**Affiliations:** 1Department of Immunology and Microbiology, Dasman Diabetes Institute, Dasman 15462, Kuwait; amal.hasan@dasmaninstitute.org (A.H.); hossein.arefanian@dasmaninstitute.org (H.A.); 2Department of Genetics and Bioinformatics, Dasman Diabetes Institute, Dasman 15462, Kuwait; arshad.channanath@dasmaninstitute.org (A.M.C.); alphonse.thangavel@dasmaninstitute.org (T.A.T.); 3Department of Biochemistry and Molecular Biology, Dasman Diabetes Institute, Dasman 15462, Kuwait; irina.alkhairi@dasmaninstitute.org (I.A.); preethi.cherian@dasmaninstitute.org (P.C.); jehad.abubakr@dasmaninstitute.org (J.A.); 4Special Services Facilities, Dasman Diabetes Institute, Dasman 15462, Kuwait; sriraman.devarajan@dasmaninstitute.org (S.D.); mohamed.abufarha@dasmaninstitute.org (M.A.-F.)

**Keywords:** COVID-19 vaccines, coagulation factors, antithrombin

## Abstract

Background: Severe coronavirus disease 2019 (COVID-19) is associated with increased risk of thrombosis and thromboembolism. Exposure to COVID-19 vaccines is also associated with immune thrombotic thrombocytopenia, ischemic stroke, intracerebral haemorrhage, and cerebral venous thrombosis, and it is linked with systemic activation of coagulation. Methods: We assess the circulating levels of coagulation factors (factors XI, XII, XIII, and prothrombin) and antithrombin in individuals who completed two doses of either ChAdOx1-S or BNT162b2 COVID-19 vaccine, within the timeframe of two months, who had no previous history of COVID-19. Results: Elevated levels of factors XI, XII, XIII, prothrombin, and antithrombin were seen compared to unvaccinated controls. Levels of coagulation factors, antithrombin, and prothrombin to antithrombin ratio were higher with BNT162b2 compared to ChAdOx1-S vaccine. Conclusions: The clinical significance of such coagulation homeostasis disruption remains to be elucidated but it is worthy of global scientific follow-up effort.

## 1. Introduction

Severe acute respiratory syndrome coronavirus-2 (SARS-CoV-2) emerged in December 2019 and caused a pandemic of coronavirus disease 2019 (COVID-19) [1]. As of 22 July 2022, there have been 565,207,160 confirmed cases worldwide, including 6,373,739 deaths [1]. Infected individuals can present with no symptoms, but most develop mild/moderate disease, whereas others succumb to severe/critical disease [2,3]. Due to the high transmissibility and pathogenicity of the virus, a worldwide effort led to the development of four vaccine platforms. These included nucleic acid (mRNA or DNA) platforms, viral vector platforms, inactivated virus platforms, and subunit vaccine platforms [4,5,6,7,8,9]. The World Health Organization and the United States Food and Drug Administration declared the release of COVID-19 vaccines in September 2020 [10]. As of 18 July 2022, a total of 12,219,375,500 vaccine doses have been administered worldwide [1].

The adenovirus-vector vaccine ChAdOx1-S (Oxford-AstraZeneca, developed by Oxford University, Oxford; and AstraZeneca, Cambridge, UK) and the lipid nanoparticle-formulated mRNA vaccine BNT162b2 (BioNTech-Pfizer, developed by BioNTech, Mainz, Germany; and Pfizer, NY, USA) are commonly used worldwide [4,5,6] and have both demonstrated effectiveness in preventing severe COVID-19 [11,12]. Severe COVID-19 is associated with increased risk of thrombosis and thromboembolism [13,14]. Studies have reported elevated D-dimer, fibrinogen, factor VIII, von Willebrand factor, and decreased antithrombin in severe COVID-19 [15]. This hypercoagulable state takes place through mechanisms unique to SARS-CoV-2 and centers around the crosstalk between thrombosis and inflammation [16]. Exposure to COVID-19 vaccines has also been associated with immune thrombotic thrombocytopenia, ischemic stroke, intracerebral hemorrhage [17], and cerebral venous thrombosis [18], and it has been linked with systemic activation of coagulation [19]. However, the effect of COVID-19 vaccines on coagulation factors and antithrombin remains to be elucidated.

The coagulation cascade consists of three pathways, these include the extrinsic, intrinsic, and common pathways, which interact together to allow for rapid healing and prevention of spontaneous bleeding. The extrinsic and intrinsic pathways originate separately but both lead into the common pathway by independently activating factor X. The intrinsic pathway utilizes factors XII (Hageman factor), XI, IX, and VIII, whereas the extrinsic pathway is initiated by factor III (tissue factor) and its interaction with factor VII. The common pathway utilizes factors X, V, II (prothrombin), I (fibrinogen), and XIII (stabilizing factor). Prothrombin is the precursor of thrombin, which is the end-product of the coagulation cascade that converts fibrinogen to a fibrin clot [20,21].

In this study, we assess the circulating levels of coagulation factors (factors XI, XII, XIII, and prothrombin) and antithrombin in individuals who completed two doses of either ChAdOx1-S or BNT162b2 COVID-19 vaccine and had no previous history of COVID-19. 

## 2. Materials and Methods

### 2.1. Study Participants

The study was conducted according to the guidelines of the Declaration of Helsinki and approved by the Ethical Review Committee of Dasman Diabetes Institute, Kuwait. All participants provided a written informed consent to participate in the study and included 166 individuals who received two doses of the ChAdOx1-S vaccine, 103 individuals who received two doses of the BNT162b2 vaccine, and 34 unvaccinated individuals (participants’ characteristics are shown in Appendix A). Peripheral blood samples (collected in EDTA tubes) were obtained within two months of receiving the second dose of either the ChAdOx1-S (1–57 days with a median of 16 days (interquartile range, IQR: 8, 29)) or the BNT162b2 (1–44 days with a median of 27 days (IQR: 18, 36)) vaccine. Samples were also obtained from unvaccinated individuals. Plasma samples were separated and stored at −80 °C until analysis.

### 2.2. Measurement of Plasma SARS-CoV-2-Specific Antibodies via Enzyme-Linked Immunosorbent Assays

Plasma levels of SARS-CoV-2-specific antibodies (IgM, IgG, and IgA) were measured using the SERION enzyme-linked immunosorbent assay (ELISA) agile SARS-CoV-2 IgA/IgG or IgM kit (SERION Diagnostics, Würzburg, Germany), as per manufacturers’ instructions. Briefly, 100 µL of controls, serum cutoffs (provided with the kit), and samples were transferred onto plates and incubated in a 37 °C moist chamber for 60 min. Following incubation, the plates were washed with 1× wash buffer, and then 100 µL of antibody-specific conjugate was added. The plates were then incubated in a 37 °C moist chamber for 30 min and washed with 1× wash buffer. Next, the chromogen TMB solution was added, and the plates were incubated in a 37 °C moist chamber for 30 min. The reaction was stopped using the stop solution and the absorbance was read using the Synergy H4 Hybrid Biotek microplate reader at an optical density (OD) of 405 nm wavelength. 

### 2.3. Measurement of Plasma Coagulation Factors and Antithrombin via Enzyme-Linked Immunosorbent Assays

Plasma coagulation factors and antithrombin were measured using the Coagulation 6-Plex Human ProcartaPlex™ Panel 1 kit (Invitrogen; Vienna, Austria), and the immunoassay was conducted as per manufacturers’ instructions. Briefly, plasma samples and standards were transferred onto plates and incubated at room temperature for 2 hours. After incubation, the plates were washed and incubated with the detection antibody cocktail for 30 min. The plates were then washed, and streptavidin-PE was added, and the plates were incubated for another 30 min. Prior to acquiring the results, the plates were washed, and the beads were resuspended in reading buffer. The plates were read using the BioPlex-200 system (Bio-Rad, Hercules, CA, USA), as per the settings specified in the kit. The results were calculated using the 5-PL nonlinear setting within the BioPlex manager software (Version 6.0; Bio-Rad, Hercules, CA, USA).

### 2.4. Statistical Analysis

Statistical analysis was conducted using R Core Team (2021), Version 4.0.5. R: A language and environment for statistical computing. R Foundation for Statistical Computing, Vienna, Austria. Continuous values were compared using Wilcoxon rank sum test. Spearman’s correlation analysis was used to analyze the correlation of duration (in days) between the second dose and coagulation factors / antithrombin. Chi-squared test or Fisher’s exact test were conducted to examine differences between groups.

## 3. Results and Discussion

Elevated levels of factors XI, XII, XIII, prothrombin, and antithrombin were observed in vaccinated individuals compared to unvaccinated controls. More specifically, the level of factors XI, XII, XIII, prothrombin, and antithrombin were significantly higher in the ChAdOx1-S (*p* = 0.001, *p* = 0.000196, *p* = 6.48, *p* = 0.000289, and *p* = 5 × 10^−7^, respectively) and BNT162b2 (*p* = 1.25 × 10^−7^, *p* = 2.43 × 10^−6^, *p* = 5.94 × 10^−7^, *p* = 6.93 × 10^−9^, and *p* = 1.57 × 10^−8^, respectively) vaccinated groups compared to the control group (Figure 1). Of note, the prothrombin to antithrombin ratio was significantly lower (*p* ≤ 0.001) in the ChAdOx1-S vaccinated group (but not the BNT162b2 vaccinated group) compared to the control group (Appendix A). In fact, the levels of factors XI, XII, XIII, prothrombin, and antithrombin (Figure 1), as well as prothrombin to antithrombin ratio (Appendix A), were significantly higher (*p* = 1.67 × 10^−7^, *p* = 0.000196, *p* = 0.000372, *p* = 6.93 × 10^−9^, and *p* = 1.46 × 10^−5^, respectively) in the BNT162b2 vaccinated group compared to the ChAdOx1-S vaccinated group (Figure 1). However, the cause of this difference is unknown and should be investigated by future studies.

Elevated levels of factors XI, XII, and XIII are known risk factors for venous thrombosis [22], hemorrhagic stroke [23], and peripheral artery disease [24], respectively. Therefore, post-vaccination increases in factors XI and XIII levels suggest a procoagulant state, wherein a concomitant increase in factor XII occurs as a counter-regulatory mechanism. Similarly, the elevated level of prothrombin, a known risk factor for deep venous and cerebral venous thromboembolism [25], was accompanied by a concomitant increase in antithrombin. Antithrombin reduces the production of thrombin from prothrombin and decreases the amount of activated factor X [26]. Therefore, the production of prothrombin may have increased as a counter-regulatory mechanism. Interestingly, biomarkers of thrombin generation or activation, including thrombin-antithrombin complex and prothrombin fragment 1.2, are markedly elevated in COVID-19 [27].

It is worthy of note that the level of factors XI, XII and XIII, prothrombin, and antithrombin decreased with time, indicating a transient procoagulant risk (Figure 2). To date, none of the participants reported a post-vaccination thrombotic event. This raises the question of whether a lack of counter-regulatory mechanisms underlie some cases of post-vaccination thrombotic events.

Because similar findings were observed with both ChAdOx1-S and BNT162b2 vaccines, one may surmise that the effects were not due to the unique formulation of either vaccine, but rather, the body’s inflammatory response to the elicited spike protein. However, there was no correlation between SARS-CoV-2-specific antibodies and coagulation factors or antithrombin in individuals vaccinated with either vaccine (Appendix A). This lack of correlation could have been due to the small number of participants in each group. In fact, when all vaccinated individuals were combined together, a direct (but weak) correlation was observed between SARS-CoV-2-specific antibodies and factor XI, prothrombin, and antithrombin (Appendix A). Similarly, when all individuals (vaccinated and unvaccinated) were analyzed together, a direct (but weak) correlation was found between SARS-CoV-2-specific antibodies and factors XI, XII, XIII, prothrombin and antithrombin (Appendix A). However, larger studies are required to ascertain/confirm whether a correlation exists between SARS-CoV-2-specific antibodies and coagulation factors and antithrombin and whether SARS-CoV-2-specific antibodies correlate with inflammatory cytokines in vaccinated individuals.

In addition, further studies are required to confirm how long coagulation factors and antithrombin remain elevated post COVID-19 vaccinations and whether these factors are also elevated after booster doses. Moreover, studies are required to elucidate the underlying molecular mechanisms that lead to these changes. Furthermore, clinical research should be conducted to investigate whether patients with medical conditions such as obesity, diabetes, autoimmune disease, and heart disease are at increased risk of thromboembolism. In the interim, coagulation factors and antithrombin may be considered when treating patients with heparin because endogenously elevated prothrombin may pose a particular risk to such patients. It is of note that our findings raise the question whether individuals with increased prothrombin to antithrombin ratio are at an increased risk of thrombosis.

Limitations of the study include: (1) different numbers of participants in each group, and small number of participants, particularly in the control group; (2) variable time duration between completion of the second dose and sample collection; (3) the presence of thrombophilic substrate (FV Leiden, FII, hyper-homocysteinemia), which could have caused a bias in the results, was not investigated. This potential bias could be addressed by measuring these factors before vaccination, after the first dose, after the second dose, a standard period after the second dose (for example 2 months), and after a booster dose, which would also show whether the observed effects are dose dependent; (4) the correlation between SARS-CoV-2-specific antibodies and inflammatory cytokines were not investigated; (5) lack of data pertaining to diet, physical activity, and behavior indicators of the participants, which could have influenced the results; (6) the socio-economic data about the participants were not collected. Nonetheless, our data shed light on an important aspect of COVID-19 vaccines, with significant clinical implications.

## 4. Conclusions

Our data show that after completing two doses of either the ChAdOx1-S or BNT162b2 vaccine, there is a transient increase in circulating levels of coagulation factors, and possibly, activation of counter-regulatory mechanisms. Larger studies are required to confirm our findings and to elucidate the molecular mechanisms as well as the clinical significance of such coagulation homeostasis disruption.

## Figures and Tables

**Figure 1 vaccines-10-01226-f001:**
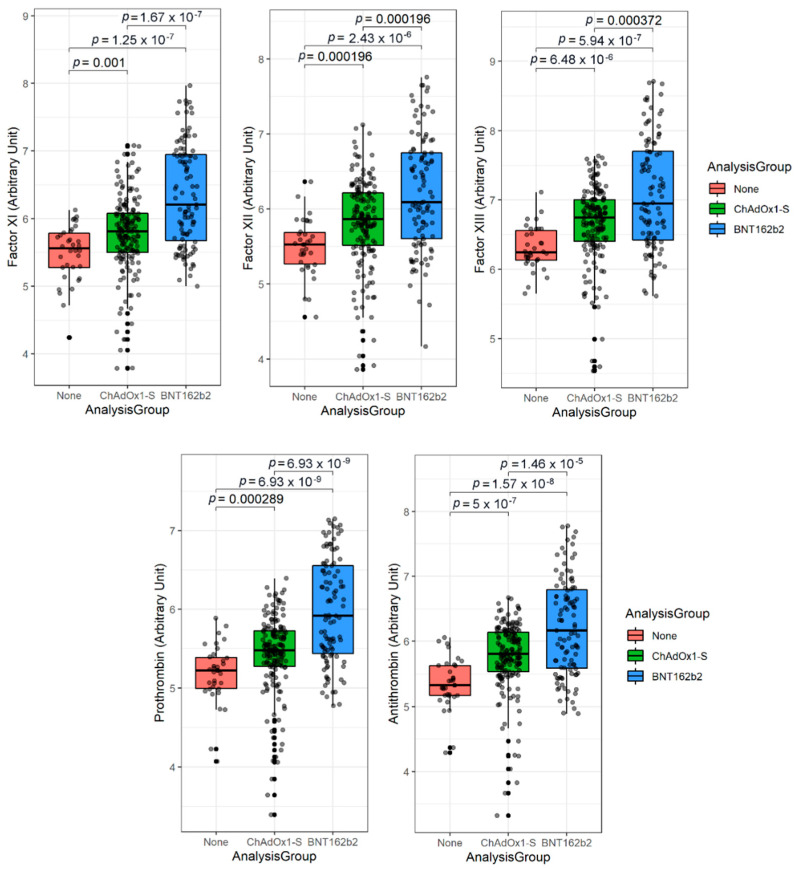
Circulating levels of coagulation factors and antithrombin after COVID-19 vaccinations. The figure shows the circulating levels of coagulation factors (XI, XII, XIII, and prothrombin) and antithrombin in the timeframe of two months since each group completed two doses of either the ChAdOx1-S (1–57 days post dose 2 with a median of 16 days (IQR: 8, 29); *n* = 166) or the BNT162b2 (1–44 days post dose 2 with a median of 27 days (IQR: 18, 36); *n* = 103) vaccine compared to a non-vaccinated group (*n* = 34). Statistical analysis was conducted using R Core Team (2021) software. Values are log transformed to reduce skewness. Continuous values were compared using the Wilcoxon rank sum test.

**Figure 2 vaccines-10-01226-f002:**
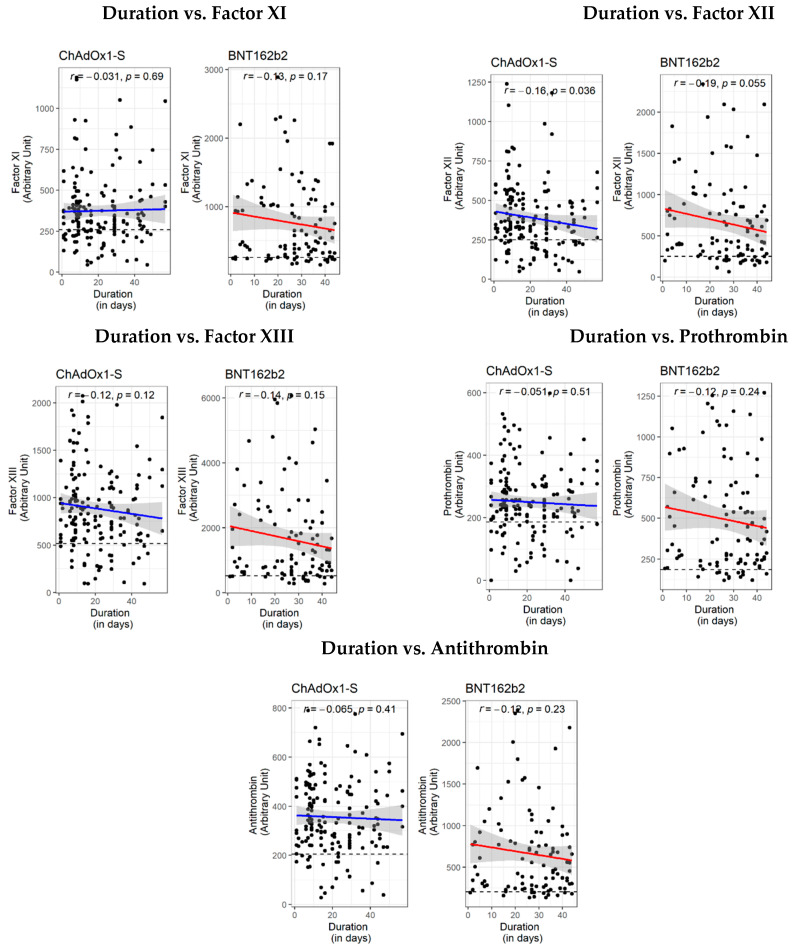
Correlation of circulating levels of coagulation factors/antithrombin with the duration (in days) between the administration of the 2nd dose of either the ChAdOx1-S or BNT162b2 vaccine and sample collection (ChAdOx1-S vaccine: 1–57 days post dose 2 with a median of 16 days (IQR: 8, 29), *n* = 166; BNT162b2 vaccine: 1–44 days post dose 2 with a median of 27 days (IQR: 18, 36), *n* = 103). The non-parametric Spearman’s *r* test was used for correlation analysis.

## Data Availability

All data generated or analyzed during this study are included in this published article.

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
