# Peer review of "Association of COVID-19 Vaccines ChAdOx1-S and BNT162b2 with Circulating Levels of Coagulation Factors and Antithrombin"

_vaccines, 2022, doi:10.3390/vaccines10081226_

Round 1

Reviewer 1 Report

The Communication Article titled "Association of COVID-19 vaccines ChAdOx1-S and BNT162b2 with circulating levels of coagulation factors and antithrombin", authored by Amal Hasan and colleagues provides evidence on the possible procoagulant or anticoagulant roles of two COVID-19 vaccines.
The study is interesting and the study design appropriate.
My Comments and Suggestions for Authors are following:

1) The authors are careful with their conclusions and I agree that the significance of their findings remains to be elucidated. However, the results are worth mentioning to drive additional research in this direction.

2) I would suggest the presentation of the data to be done in a logarithmic scale, in order the significant differences to be better elucidated.

Author Response

Reviewer 1

The Communication Article titled "Association of COVID-19 vaccines ChAdOx1-S and BNT162b2 with circulating levels of coagulation factors and antithrombin", authored by Amal Hasan and colleagues provides evidence on the possible procoagulant or anticoagulant roles of two COVID-19 vaccines.
The study is interesting and the study design appropriate.

My Comments and Suggestions for Authors are following:

1) The authors are careful with their conclusions and I agree that the significance of their findings remains to be elucidated. However, the results are worth mentioning to drive additional research in this direction.

We hope that further research will be conducted to elucidate the significance of our findings.  

2) I would suggest the presentation of the data to be done in a logarithmic scale, in order the significant differences to be better elucidated.

We agree with the reviewer; the Boxplots of coagulation factors are now presented in logarithmic scale (Figure 1).

In addition, the English language and style has been improved, and the spelling checked - changes are highlighted throughout the manuscript.

Reviewer 2 Report

This study aims to assess the circulating levels of coagulation factors (factors XII, XI, prothrombin, XIII) and antithrombin in subjects with a negative history of SARS-CoV-2 infection, monitoring them for two months after two doses of either ChAdOx1-S or BNT162b2 COVID-19.

Despite the study could be of interest, several important limitations are present.

1- The number of the enrolled subjects is too different in the three groups (particularly the control group is composed of 34 controls).

2- The authors missed to investigate the presence of thrombophilic substrate (FV Leiden, FII, hyperhomocysteinemia): these data could influence their conclusion. For example, in order to avoid this bias, the authors could be monitored the same subjects before vaccination (T0), after the first dose (T1), after the second dose (T2), and after a standard period after the second dose (for example, 2 months later).

3- The authors missed to declare the number of days after the vaccine administration: this is an important aspect;

4- the authors missed to insert the socio-economic data about the enrolled subjects;

5- the authors missed to collect data about several important factors that could influence their data (i.e. diet, physical activities, etc...)

Author Response

Reviewer 2

This study aims to assess the circulating levels of coagulation factors (factors XII, XI, prothrombin, XIII) and antithrombin in subjects with a negative history of SARS-CoV-2 infection, monitoring them for two months after two doses of either ChAdOx1-S or BNT162b2 COVID-19.

Despite the study could be of interest, several important limitations are present.

1- The number of the enrolled subjects is too different in the three groups (particularly the control group is composed of 34 controls).

We acknowledge that the number of enrolled subjects is too different among the three groups, and this has now been mentioned in the manuscript as a limitation of the study, as follows:

‘(1) different numbers of participants in each group, and low number (n=34) of participants in the control group’ (page 3, line 141).

2- The authors missed to investigate the presence of thrombophilic substrate (FV Leiden, FII, hyperhomocysteinemia): these data could influence their conclusion. For example, in order to avoid this bias, the authors could be monitored the same subjects before vaccination (T0), after the first dose (T1), after the second dose (T2), and after a standard period after the second dose (for example, 2 months later).

We agree with the reviewer, however, the presence of thrombophilic substrate (FV Leiden, FII, hyperhomocysteinemia) was not measured. Unfortunately, we do not have samples from these participants before vaccination. However, we could recruit individuals who are still unvaccinated, and obtain samples before and after vaccination as described. Nonetheless, we have addressed this limitation in the manuscript, as follows:

‘(3) the presence of thrombophilic substrate (FV Leiden, FII, hyper-homocysteinemia) was not investigated, which could have caused a bias in the results. This potential bias could be addressed by measuring these factors before vaccination, after the first dose, after the second dose, after a standard period after the second dose (for example, 2 months), and after a booster. This will also show whether the observed effects are dose dependent’ (page 3, line 145).

3- The authors missed to declare the number of days after the vaccine administration: this is an important aspect.

We agree that this is an important aspect, which is why we analysed the correlation of duration between the second dose and test (Figure 2). This showed a decline in coagulation factor levels with time, indicating a transient procoagulant risk: ‘It is worthy of note that the level of factors XII, XI, XIII, prothrombin, and antithrombin decreased with time, indicating a transient procoagulant risk (page 3, line 113; Figure 2). The median of the number of days after vaccine administration was shown in the supplemental material S1-S3; however, we agree that this information should be specified in the manuscript (including the number of days). In this regard, the specific time frame for the ChAdOx1-S vaccine was 1-57 days post dose 2 (n=166; median 16, IQR (8, 29)), and for the BNT162b1 vaccine was 1-44 days post dose 2 (n=103; median 27, IQR (18, 36)). This information has now been added to the manuscript, as follows:

‘The samples and related data of consenting participants were obtained, and included 166 individuals who received two doses of the ChAdOx1-S vaccine (1-57 days post dose 2 with a median of 16 (IQR 8, 29); AstraZeneca-Oxford), 103 individuals who received two doses of the BNT162b2 vaccine (1-44 days post dose 2 with a median of 27 (IQR 18, 36); Pfizer-BioNTech), and 34 unvaccinated controls’ (page 2, line 73).

The same modifications were also made on page 4 in figure 1 legend, and page 5 in figure 2 legend.   

4- the authors missed to insert the socio-economic data about the enrolled subjects;

We agree that it would have been informative to insert the socio-economic data, however, we did not collect these data from the participants. Nonetheless, we have added this limitation, as follows:

‘(6) the socio-economic data about the participants was not collected’ (page 4, line 153).

5- the authors missed to collect data about several important factors that could influence their data (i.e. diet, physical activities, etc...)

We agree with the reviewer, however, this data was not collected from the participants. Nonetheless, we have added this limitation, as follows:

(5) lack of data pertaining to diet, physical activity, and behaviour indicators of the participants, which could have influenced the results (page 4, line 152).  

In addition, the English language and style has been improved, and the spelling checked - changes are highlighted throughout the manuscript.

Reviewer 3 Report

The findings appear to be interesting. Specific points that the authors need to address are as follows:

1. It is not clear that why a timeframe of two months since completing two doses of vaccines was selected.Was the observed response dose or time-dependent.

2. The response to booster doses of two selected vaccines should also be evaluated.

3. The molecular mechanisms that contribute to elevated levels of antithrombin and factors XII, XI, prothrombin, and XIII should be analyzed.

4. It is also not clear that why the levels of coagulation factors, as well as prothrombin to antithrombin ratio were higher with BNT162b2 compared to ChAdOx1-S vaccine.

5. Proper statistical analysis should be conducted for all the figures.

6. Typographical errors were found throughout the manuscript and should be corrected.

Reviewer 4 Report

The article entitled “Association of COVID-19 vaccines ChAdOx1-S and BNT162b2 with circulating levels of coagulation factors and antithrombin” focuses on the inflammatory state in vaccinated subjects with two doses of AstraZeneca Oxford or Pfizer-BioN-Tech compared to unvaccinated subjects. This topic may be interesting as the inflammation can cause  thrombosis.

The authors only studied coagulative factors such as factor XII, factor XI, prothrombin and factor XIII and antithrombin and found higher levels of coagulation factors and antithrombin in vaccinated subjects with Pfizer-BioN-Tech. This study has some weaknesses.

The authors state that the individual inflamatory response causes the inflammation. This conclusion requie the correlation of SARS-CoV-2 specific antibodies with the coagulation factors and antithrombin and the measurement of inflammatory cytokines and correlation with SARS-CoV-2 specific antibodies. Therefore, I recommend these major recommendations. Therefore, this article is nor suitable for publication in in its current version.

Author Response

Reviewer 4

The article entitled “Association of COVID-19 vaccines ChAdOx1-S and BNT162b2 with circulating levels of coagulation factors and antithrombin” focuses on the inflammatory state in vaccinated subjects with two doses of AstraZeneca Oxford or Pfizer-BioN-Tech compared to unvaccinated subjects. This topic may be interesting as the inflammation can cause  thrombosis.

The authors only studied coagulative factors such as factor XII, factor XI, prothrombin and factor XIII and antithrombin and found higher levels of coagulation factors and antithrombin in vaccinated subjects with Pfizer-BioN-Tech. This study has some weaknesses.

The authors state that the individual inflamatory response causes the inflammation. This conclusion requie the correlation of SARS-CoV-2 specific antibodies with the coagulation factors and antithrombin and the measurement of inflammatory cytokines and correlation with SARS-CoV-2 specific antibodies. Therefore, I recommend these major recommendations. Therefore, this article is nor suitable for publication in in its current version.

We agree with the reviewer, and accordingly, we have assessed the correlation between SARS-CoV-2 specific antibodies and coagulation factors and antithrombin. We also agree with the reviewer that it is essential that the correlation between inflammatory cytokines and SARS-CoV-2 specific antibodies are investigated. Together, we have addressed this in the manuscript as follows:

‘Since similar findings were observed with both vaccines, one may surmise that the ef-fects were not due to the unique formulation of either vaccine, but rather, the body’s inflammatory response to the elicited spike protein. However, there was no correlation between SARS-CoV-2-specific antibodies and coagulation factors or antithrombin in individuals vaccinated with either vaccine (supplementary material: Tables S5 & S6). This lack of correlation could have been due to the low number of participants in each group. In fact, when all vaccinated individuals were combined together, a direct (but weak) correlation was found between SARS-CoV-2-specific antibodies and factor XI, prothrombin, and antithrombin (supplementary material: Table S7). Similarly, when all individuals (vaccinated and unvaccinated) were analyzed together, a direct (but weak) correlation was found between SARS-CoV-2-specific antibodies and factors XI, XII, XIII, prothrombin and antithrombin (Supplementary material: Table S8). However, larger studies are required to ascertain/confirm whether a correlation exists between SARS-CoV-2-specific antibodies and coagulation factors and antithrombin, and whether SARS-CoV-2-specific antibodies correlate with inflammatory cytokines in vaccinated individuals’ (page 3, line 118).

In addition, the English language and style has been improved, and the spelling checked - changes are highlighted throughout the manuscript.

Round 2

Reviewer 2 Report

The authors have addressed my concerns partially, inserting several observations as the study's limitations. 

Author Response

We found the reviewer’s comments very useful, which led to a substantial improvement of the manuscript. We do regret that we could not address some of the concerns by conducting more tests or recruiting more participants. In addition, we could not change the design of the study at this stage. However, these limitations were addressed in the manuscript.

We have improved other aspects of the manuscript further as per reviewers recommendations, as follows (kindly see attached revised manuscript):  

  • We have provided more background in the introduction, included more references, and revised other references for relevance.
  • We have improved the methods section by adding more details.
  • We have improved the results section by adding more details/better description, and presenting the data in the figures in a logarithmic scale.
  • We have downplayed the conclusions as per limitations of the study

We hope that the revised manuscript will now be acceptable for publication. 

Reviewer 3 Report

The authors have addressed my concerns.

Author Response

Thank you for your review which has greatly improved the manuscript.